# Isolation and Characterization of Clinical RSV Isolates in Belgium during the Winters of 2016–2018

**DOI:** 10.3390/v11111031

**Published:** 2019-11-06

**Authors:** Winke Van der Gucht, Kim Stobbelaar, Matthias Govaerts, Thomas Mangodt, Cyril Barbezange, Annelies Leemans, Benedicte De Winter, Steven Van Gucht, Guy Caljon, Louis Maes, Jozef De Dooy, Philippe Jorens, Annemieke Smet, Paul Cos, Stijn Verhulst, Peter L. Delputte

**Affiliations:** 1Laboratory of Microbiology, Parasitology and Hygiene, and Infla-Med Centre of Excellence, University of Antwerp (UA), Universiteitsplein 1 S.7, 2610 Antwerp, Belgium; winke.vandergucht@uantwerpen.be (W.V.d.G.); Kim.stobbelaar@uantwerpen.be (K.S.); Matthias.govaerts@uantwerpen.be (M.G.); annelies.leemans@uantwerpen.be (A.L.); guy.caljon@uantwerpen.be (G.C.); louis.maes@uantwerpen.be (L.M.); paul.cos@uantwerpen.be (P.C.); 2Pediatrics Department, Antwerp University Hospital (UZA), Wilrijkstraat 10, 2650 Edegem, Belgium; Thomas.mangodt@uza.be (T.M.); Stijn.verhulst@uza.be (S.V.); 3Sciensano, Rue Juliette Wytsmanstraat 14, 1050 Brussels, Belgium; Cyril.barbezange@sciensano.be (C.B.); steven.vangucht@sciensano.be (S.V.G.); 4Laboratory of Experimental Medicine and Pediatrics, University of Antwerp (UA), Univeristeitsplein 1 T.3, 2610 Antwerp, Belgium; benedicte.dewinter@uantwerpen.be (B.D.W.); Jozef.dedooy@uza.be (J.D.D.); Philippe.jorens@uza.be (P.J.); annemieke.smet@uantwerpen.be (A.S.); 5Pediatric intensive care unit, Antwerp University Hospital (UZA), Wilrijkstraat 10, 2650 Edegem, Belgium

**Keywords:** patient-derived virus, RSV, bronchiolitis, mucin

## Abstract

Respiratory Syncytial Virus (RSV) is a very important viral pathogen in children, immunocompromised and cardiopulmonary diseased patients and the elderly. Most of the published research with RSV was performed on RSV Long and RSV A2, isolated in 1956 and 1961, yet recent RSV isolates differ from these prototype strains. Additionally, these viruses have been serially passaged in cell culture, which may result in adaptations that affect virus–host interactions. We have isolated RSV from mucosal secretions of 12 patients in the winters 2016–2017 and 2017–2018, of which eight RSV-A subtypes and four RSV-B subtypes. Passage 3 of the isolates was assessed for viral replication kinetics and infectious virus production in HEp-2, A549 and BEAS-2B cells, thermal stability at 37 °C, 32 °C and 4 °C, syncytia formation, neutralization by palivizumab and mucin mRNA expression in infected A549 cells. We observed that viruses isolated in one RSV season show differences on the tested assays. Furthermore, comparison with RSV A2 and RSV B1 reveals for some RSV isolates differences in viral replication kinetics, thermal stability and fusion capacity. Major differences are, however, not observed and differences between the recent isolates and reference strains is, overall, similar to the observed variation in between the recent isolates. One clinical isolate (BE/ANT-A11/17) replicated very efficiently in all cell lines, and remarkably, even better than RSV A2 in the HEp-2 cell line.

## 1. Introduction

Respiratory Syncytial Virus (RSV), recently renamed to human orthopneumovirus, is a very important viral respiratory pathogen in infants, children and adult patients with immunodeficiency or cardiopulmonary disease and it is recognized as a major threat for the elderly population [1,2,3,4,5,6]. An RSV infection starts with typical common-cold like symptoms but may progress to serious lower respiratory tract infections associated with a high rate of hospitalization of infants, children and elderly [7]. No vaccines or therapeutics are available except for Synagis^®®^, also known as the humanized antibody palivizumab. Palivizumab, which is solely used for passive immunization of high-risk infants, targets a specific, highly conserved epitope on the fusion protein [8,9], resulting in fusion inhibition. Currently, treatment of severe lower respiratory tract infections as a result of RSV infection consists of supportive care only, such as oxygen administration and nutrition.

RSV is classified in the family Pneumoviridae, genus *Orthopneumovirus* and can be divided into two subtypes, RSV-A and RSV-B. It has a non-segmented, negative, single-stranded RNA genome that consists of ten genes, encoding 11 proteins. The viral envelope contains three proteins: the attachment protein (G), the fusion protein (F) and the small hydrophobic protein (SH). The G protein interacts with cellular receptors on the host cell membrane to attach the virus particle to the cell surface. The protein consists of a central conserved domain, two glycosylated mucin-like regions and an N-terminal region containing a transmembrane domain and a cytoplasmic domain [10,11]. Sequencing of the G gene indicated that the two mucin-like regions flanking the central domain only have a 67% similarity at the nucleotide level between RSV-A and RSV-B and only 53% similarity at the deduced amino acid levels [12]. Consequently, the two mucin-like regions serve as excellent targets for RSV evolution studies. Both subtypes are further divided into genotypes based on those genetic variations. For RSV-A, the genotypes GA1-7, SAA1-2, NA1-4 and ON1 [13,14,15,16,17,18] have been defined, while for RSV-B, the GB1-5, SAB1-4, URU1-2, BA1-12 and THB [13,14,19,20,21,22,23,24,25] genotypes are reported. The F protein is responsible for the fusion of the viral envelope with the host cell membrane. An important side effect is the fusion of the cell membranes of an infected cell with adjacent cells, resulting in a giant cell with multiple nuclei, better known as a syncytium [26]. The formation of syncytia is recognized as a means to efficiently spread the infection along epithelial surfaces, while minimizing contact with the immune system [27].

One of the hallmarks of the pathology caused by RSV infection is increased mucus production in the lungs of infected individuals. Mucus is a gel-like substance that consists of different mucins (MUC), which are high molecular mass, highly glycosylated glycoproteins [28]. Airway mucus protects the epithelial surface from injury through mucociliary clearance, facilitating the removal of foreign particles and chemicals that enter the lung. Twenty-one MUC proteins have been described in humans and are divided in two families: secreted mucins and cell-tethered mucins. The major mucins produced in the airways are MUC5AC and MUC5B as secreted mucins and MUC1, MUC4, MUC16 and MUC20 as membrane-bound mucins [29].

Most of the published research on RSV was performed using the laboratory strains RSV-Long and RSV A2, which were isolated from the population in 1956 and 1961, respectively [30,31]. Not only have these viruses not circulated in the public for many years, they have been serially passaged in cell culture, which may have resulted in an accumulation of mutations that benefit the virus to grow in cell culture but may also affect virus-host interactions. Low-passage, characterized clinical strains are hard to come by and consequently less used in research. Therefore, we have isolated virus from mucosal secretions of 12 patients in the winter seasons of 2016–2017 and 2017–2018 in Belgium, resulting in eight RSV-A subtypes and four RSV-B subtypes. We have grown these viruses to passage 3 and used them to assess their viral replication kinetics and infectious virus production in HEp-2, A549 and BEAS-2B cells, thermal stability at 37 °C, 32 °C and 4 °C, syncytium formation and neutralization by palivizumab. We have also obtained G protein sequences to assign genotypes and evaluated production of mucin mRNA expression in A549 cells upon infection.

## 2. Methods

### 2.1. Cells and Viruses

The HEp-2, A549 and Vero cell lines were obtained from and cultured to the instructions of ATCC. The BEAS-2B cell line was a generous gift from Dr. Ultan F. Power (Queens University Belfast, Ireland). All cells were cultured in Dulbecco’s modified Eagle medium containing 10% inactivated fetal bovine serum (DMEM-10) (Thermo Fisher Scientific, Waltham, MA USA). RSV reference strains A2 and B1 were obtained from BEI resources, RSV A2 was cultivated in HEp-2 cells as described by Van der Gucht W. et al. [32] and RSV B1 was cultivated on Vero cells in medium containing 2% inactivated foetal bovine serum (iFBS) until cytopathic effects (CPE) were visible throughout the flask. Virus was collected as described for RSV A2 and quantified in a conventional plaque assay on HEp-2 cells as described by Schepens et al. [33]. Briefly, HEp-2 cells were seeded at a concentration of 17,500 cells/well in clear 96 well plates (Falcon) one day prior to infection. Cells were washed with DMEM without iFBS (DMEM-0) and infected with 50 µL of a 1/10 dilution series made in DMEM-0. Cells were incubated for 2h at 37 °C after which the inoculum was replaced by DMEM-10 containing 0.6% Avicel^®®^ (FMC biopolymer) and incubated for three additional days at 37 °C (5% CO_2_). Afterwards, cells were washed with PBS, fixed with 4% paraformaldehyde solution and stained with palivizumab (leftovers provided by the Department of Paediatrics, Antwerp University Hospital) and goat-anti human secondary IgG conjugated with horseradish peroxidase (HRP) (Thermo Fisher Scientific) and visualized using chloronaphtol solution (Thermo Fisher Scientific).

### 2.2. Virus Isolation from Clinical Samples

This study was approved by the ethical committee of the Antwerp University Hospital and the University of Antwerp (16/46/491). Nasal secretions were collected from children showing symptoms of an RSV-related bronchiolitis during the winter seasons of 2016–2017 and 2017–2018 after written parental consent was given. The secretions were extracted by a nasal swab and/or a nasopharyngeal aspirate, which were stored at 4 °C for less than 10 h. One day prior to mucosal extraction, HEp-2 cells were seeded at a concentration of 17,500 cells/well in a clear 96 well plate (Falcon, Corning, NY, USA). Samples were vortexed for one minute with glass beads (Sigma-Aldrich; St. Louis, Mo, USA) before inoculating HEp-2 cells with 50 µL of a ¼ dilution series of the sample, made in DMEM-0. After 2 h of incubation with the inoculum at 37 °C, 50 µL of DMEM containing iFBS, antibiotics (penicillin/streptomycin (Thermo Fischer Scientific), moxifloxacin (Sigma-Aldrich)) and anti-fungals (Fungizone)(Sigma-Aldrich) was added to obtain a final concentration of DMEM with 2% FBS. Plates were incubated for seven days at 37 °C (5% CO_2_). After seven days, the plates were checked for syncytium formation and 50 µL of the well with the lowest concentration of original sample but still presenting CPE, was transferred to a newly seeded plate, following the previously described protocol. After an additional seven days, the wells were rechecked for syncytium formation. A total of 250 µL from wells presenting with syncytia was transferred to a freshly seeded T25, which was incubated until CPE was visible throughout the culture. Supernatant was collected, centrifuged for ten minutes at 1000× *g*, aliquoted, and snap frozen in liquid nitrogen (passage 0). Virus obtained from these clinical samples was propagated until passage 3 on HEp-2 cells to obtain a plaque forming unit (PFU) count high enough to perform the following experiments. One sample did not propagate efficiently on HEp-2 cells and was propagated for three passages on Vero cells until a high PFU was reached. Passage 3 cultures were tested for the following contaminants by qPCR: hMPV, Rhinovirus 1 and 3, PIV 1, 2, 3 and 4, adenovirus, Coronavirus 229E, NL63, OC43, Paraechovirus, Enterovirus 68 and Bocavirus and found negative for all. Several virus cultures tested negative on mycoplasma presence and no signs of bacterial or fungal presence was observed when the cultures were test grown in absence of antibiotics and anti-fungals.

### 2.3. RSV-A and RSV-B Subtyping

RNA for subtyping was extracted from passage 0 virus using the QIamp viral RNA extraction mini kit (QIAgen, Hilden, Germany) following the instructions of the manufacturer. A multiplex reaction mix was made with superscript III platinum one-step quantitative kit (Thermo Fisher Scientific) in a final volume of 25 µL containing 5 µL RNA, 12.5µL PCR master mix, 1 µL superscript RT/Platinum Taq polymerase and 2.5 µL of a pre-mixed primer/probe solution. This solution contains a final concentration of 5 µM of each primer and 1 µM of each probe. The primers for RSV-A are located in the L gene (RSVQA1: 5′ - GCT CTT AGC AAA GTC AAG TTG AAT GA–3′ and RSVQA2: 5′–TGC TCC GTT GGA TGG TGT AAT–3′, RSVQA probe: 5′–HEX/ACA CTC AAC/ZEN/AAA GAT CAA CTT CTG TCA TCC AGC - ‘3 - IABkFQ) and the primers for RSV-B are located in the N gene (RSVQB1: 5′–GAT GGC TCT TAG CAA AGT CAA GTT AA–3′ and RSVQB2: 5′–TGT CAA TAT TAT CTC CTG TAC TAC GTT GAA–3′, RSVQB probe: 5′–RTEX615/TGA TAC ATT AAA TAA GGA TCA GCT GCT GTC ATC CA–‘3 - BHQ_2) (adapted from Bragstad K. [34]). The reaction was run on a Real-time PCR machine (Stratagene, Mx3000P, Thermo Fisher Scientific) with the following program: 50 °C for 30 min, 94 °C for 5 min followed by 45 cycles of 15 s at 94 °C and 1 min at 55 °C. Ct values below 40 were counted as positive.

### 2.4. Gene Nucleotide Sequencing and Phylogenetic Analysis

Viral RNA was extracted from infected cell culture supernatants using the QIAmp viral RNA mini kit (Qiagen) according to the instructions provided by the manufacturer. Viral RNA of the G gene was transcribed to cDNA and amplified using the One-step RT-PCR kit (Qiagen) and the following primers as described by Houspie et al. [35]. For RSV-A, G267FW: 5′–ATG CAA CAA GCC AGA TCA AG–3′ and F164RV: 5′–GTT ATC ACA CTG GTA TAC CAA CC - 3′, for RSV-B, BGF: 5′–GCA GCC ATA ATA TTC ATC ATC TCT–3′ and BGR: 5′–TGC CCC AGR TTT AAT TTC GTT C–3′ were used. Primers were added to the reaction mix consisting of 10 µL 5× RT-PCR buffer, 2 µL dNTP, 2 µL enzyme, 20 µL H2O to a final amount of 30 pmol. Ten microliters (10 µL) of RNA extract was added to the reaction mix. The PCR was performed in a thermocycler (Unocycler, VWR; Radnor, PA, USA) with the following program: 30 min at 50 °C for the RT step, 15 min at 95 °C for PCR activation, 40 amplification cycles consisting of 30 s at 95 °C, 1 min at 55 °C and 1 min at 72 °C followed by a final extension step of 10 min at 72 °C. The length of the amplified cDNA was verified with 1% agarose gel electrophoresis and cDNA was visualized with Gelgreen™ (VWR). Amplified cDNA was delivered to the VIB Neuromics support facility (University of Antwerp) for PCR cleanup and DNA sequencing with the following primers as described by Houspie et al. [35]: in addition to the PCR amplification primers, for RSV-A: G516R (5′ - GCT GCA GGG TAC AAA GTT GAA C–3′) and G284F (5′–ACC TGA CCC AGA ATC CCC AG–3′) and for RSV-B: BGF3 (5′–AGA GAC CCA AAA ACA CYA GCC AA–3′) and BGR3 (5′–ACA GGG AAC GAA GTT GAA CAC TTC A–3′) were provided for sequencing.

Sequences were annotated in SnapGene and contigs were built in BioEdit with the CAP3 application. Multiple sequence alignments from reference strains and contigs, and phylogenetic trees were constructed in MEGA X using the maximum likelihood method.

### 2.5. Viral Replication Kinetics

HEp-2, A549 and BEAS-2B cells were seeded at a concentration of 17,500 cells/well in black CELLSTAR^®®^ 96 well plates with a µclear^®®^ flat bottom suitable for fluorescence microscopy (Greiner-bio one) one day prior to inoculation. Briefly before inoculation, the cells were washed with DMEM-0. Clinical RSV and RSV A2 were diluted to infect the cells at a multiplicity of infection (MOI) of 0.01. Virus was left to adhere for 2 h at 37 °C, 5% CO_2_ and replaced with DMEM-10. Cells were fixed with 4% paraformaldehyde after 24 h, 48 h and 72 h, permeabilized with triton X-100 and stained with polyclonal goat-anti-RSV antibody (Virostat; specificity all viral antigens) followed by donkey-anti-goat secondary antibody conjugated with Alexa Fluor 488 (AF488) (Thermo Fisher Scientific) and additional DAPI nucleus staining (Sigma-Aldrich).

### 2.6. Infectious Virus Production

HEp-2, A549 and BEAS-2B cells were seeded at a concentration of 200,000 cells/well in 24-well plates 24 h prior to infection. Briefly, before infection, cells were washed with DMEM-0 and afterwards infected with clinical isolates and RSV A2 and RSV B1 at a MOI of 0.01. The supernatant was collected after 24 h, 48 h and 72 h, aliquoted, snap frozen and stored at –80 °C. The supernatant was quantified using a conventional plaque assay on HEp-2 cells as described above.

### 2.7. Thermal Stability Assay

Aliquots of clinical isolates and RSV A2 and RSV B1 were thawed, diluted in DMEM-0 to obtain a starting concentration of 1 × 10^5^ PFU/mL and aliquots were made of each sample. Immediately afterwards, one aliquot of each sample was snap frozen in liquid nitrogen as T0. The other aliquots were stored at 4 °C, at 32 °C or at 37 °C for 24 h, 48 h and 72 h, snap frozen in liquid nitrogen and stored at –80 °C until quantification was performed. A conventional plaque assay on HEp-2 cells, as described above, was used to quantify the remaining PFU in each aliquot.

### 2.8. Cell-to-Cell Fusion Assay

One day prior to inoculation, HEp-2 cells were seeded at a concentration of 17,500 cells/well in black CELLSTAR^®®^ 96-well plates with a µclear^®®^ flat bottom suitable for fluorescence microscopy (Greiner bio-one). Cells were inoculated with clinical RSV and RSV A2 at a MOI of 0.05 for 2 h at 37 °C (5% CO_2_). After 2h, the inoculum was removed and replaced by DMEM-10 containing 0.6% Avicel^®®^ (FMC biopolymer). After 48 h cells were washed with PBS, fixed with 4% paraformaldehyde solution, permeabilized with triton X-100 and stained with polyclonal goat-anti-RSV (Virostat; Westbrook, ME, USA) followed by donkey-anti-goat secondary antibody conjugated with AF488 (Thermo Fisher Scientific). DAPI staining was performed to stain the nuclei (Sigma-Aldrich).

### 2.9. Plaque Reduction Assay

The plaque reduction assay was performed as described by Leemans A. et al. [36]. Briefly, HEp-2 cells were seeded at a concentration of 17,500 cells/well in a clear 96-well plate (Falcon) 24 h prior to inoculation. Palivizumab was diluted 1/40 and further in a ½ dilution series, which was incubated with diluted virus for 1 h at 37 °C (5% CO_2_). Afterwards, the cells were washed briefly with DMEM-0, and inoculated with 50 µL of the virus-antibody solution for 2 h at 37 °C (5% CO_2_). Then, the inoculum was replaced with DMEM-10 containing 0.6% Avicel^®®^ (FMC biopolymer). The plates were incubated for three days at 37 °C (5% CO_2_), washed with PBS and fixed with 4% paraformaldehyde solution. The cells were permeabilized with triton X-100, stained with palivizumab antibody followed by goat-anti-human IgG conjugated with HRP and coloured using chloronaphtol solution (Thermo Fisher Scientific). The neutralization titer was defined as the reciprocal of the highest antibody dilution producing 50% reduction in plaques (ED50), relative to virus-control wells without antibody.

### 2.10. Mucin mRNA Expression Assay

A549 cells were seeded at a concentration of 200,000 cells/well in 24-well plates, 24 h prior to inoculation (Greiner bio-one). Cells were infected with a MOI of 0.1 for 2 h at 37 °C (5% CO_2_). After 2 h, inoculum was replaced by DMEM-10 and was incubated for an additional 48 h. Afterwards, cell supernatant was collected, spun down at 1000× *g* for 15 min and only the pellet was kept. The still adherent cells were lysed with lysis buffer from the nucleospin kit (Macherey-Nagel; Düren, Germany) and added to the pellet. The solution was thoroughly mixed and frozen at –80 °C until extraction was performed. RNA isolation was done following manufacturer’s instructions of the nucleospin RNA kit (Macherey-Nagel). Concentrations were evaluated using the Nanodrop^®®^ (Thermo Fisher Scientific) and 1 µg of RNA was used to convert to cDNA using the SensiFast™ cDNA synthesis kit (Bioline; London, UK). Relative gene expression was determined with the GoTaq qPCR master mix (Promega; Madison, WI, USA) with SYBR Green Fluorescence detection on a QuantStudio 3 Real-time PCR instrument (Thermo Fisher Scientific). Standard QuantiTect primers available from Qiagen were used for GAPDH (QT00079247), ß-actin (QT00095431), MUC1 (QT00015379), MUC4 (QT00045479), MUC5AC (QT00088991) and MUC5B (QT01322818). Analysis and quality control were performed using qbase+ software (Biogazelle; Ghent, Belgium), relative expression of the target genes was normalized to the expression of the housekeeping genes GAPDH and ß-actin.

### 2.11. Fluorescence Microscopy and Image Analysis

Fluorescence images were acquired using an Axio Observer inverted microscope and a Compact Light source HXP 120C with filter sets 49, 10 and 20 for blue, green and red fluorophores, respectively (Zeiss). Image analysis was done using Zeiss ZEN 2.3 blue edition imaging software and ImageJ version 2.0.0-rc-43/1.50e. Calculations were made in Excel for Mac and Graphpad Prism 6.

### 2.12. Statistical Analysis

Data for viral replication kinetics, infectious virus production and thermal stability are presented as means (± SEM) of the indicated independent repeats. To determine the significance between the clinical isolates and the reference (RSV A2 or B1), data was analysed with a two-way ANOVA. Fusion data and MUC expression represents means (± SEM), significance was calculated between the clinical isolates and their references with a one-way ANOVA. Data for plaque reduction represents means (± SD), significance was calculated between clinical isolates and references with a one-way ANOVA. Calculations were done using Graphpad Prism 6.

## 3. Results

### 3.1. Clinical Samples and Detection of RSV

Nasal swabs and nasopharyngeal aspirates were obtained from one patient in December of 2016 and from 24 patients between October and January 2017–2018. RSV-A was detected in one sample of December 2016 and in 11 samples collected between October 2017 and January 2018. RSV-B was detected in 11 samples collected between October 2017 and January 2018. Of the remaining two RSV-negative samples, one tested positive for human metapneumovirus (hMPV), one remained negative for RSV, hMPV and Rhinovirus 1. HEp-2 cells were infected with the samples on the day of the aspiration of secretions or the day afterwards, without freezing the samples. After two weeks of incubation, 11 samples did not result in syncytium formation or positive fluorescent staining in either the nasal swab culture or the aspirate culture and were, therefore, not used in any of the following assays. Cultures that showed syncytium formation were used to grow the virus on HEp-2 cells. One sample was further grown on Vero cells since no significant titers could be reached growing the virus on HEp-2 cells (Table 1).

### 3.2. Phylogenetic Analysis

Sequences of the G gene of all samples were obtained and aligned with previously reported representative sequences from GenBank. The phylogenetic trees of RSV-A and RSV-B sequences are shown in Figure 1A,B.

All RSV-A sequences cluster within the ON1 genotype that contains a 72nt duplication and all RSV-B sequences contain a 60nt duplication in the G-gene, assigning them to the BA genotype, further differentiated into the BAIX genotype.

### 3.3. Viral Replication Kinetics

To study the dynamics of virus infection, both viral replication kinetics and infectious virus production were assessed in HEp-2, A549 and BEAS-2B cells. Cells were infected for 24 h, 48 h and 72 h with a MOI of 0.01, fixed, fluorescently stained and analysed with fluorescence microscopy to evaluate viral replication kinetics. Infectious virus production was evaluated through the collection of supernatants after 24 h, 48 h and 72 h post-infection with a MOI of 0.01. Supernatant was snap frozen and used for quantification via plaque assay. Viral replication kinetics in HEp-2 cells for RSV-A (Figure 2A) strains yielded one strain (BE/ANT-A11/17) that resulted in significantly higher percentages of RSV-infected cells after 48 h compared to RSV A2. The BE/ANT-A11/17 also produced more infectious virus particles after 24 h post inoculation (p.i.) compared to all other strains (Figure 2C). Three strains (BE/ANT-A21/17, BE/ANT-A7/17, BE/ANT-A8/17) replicated more slowly than the RSV A2 at 48 h but a fully infected culture was observed after 72 h of infection. Infection with a MOI of 0.01 of the RSV-B strains resulted in infection values between 0.5% and 1% infected cells after 24 h. The RSV-B strains (Figure 2B,D) showed two strains grown on HEp-2 cells (BE/ANT-B13/17, BE/ANT-B15/17) and one strain grown on Vero cells (BE/ANT-B20/17) that resulted in significantly more infected cells at 72 h than the reference RSV B1, whereas just one strain (BE/ANT-B2/17) seemed to result in comparable infection as the RSV B1. Infectious virus production of RSV-B strains shows that even though the BE/ANT-B20/17 and BE/ANT-B15/17 reach a very high percentage of infected cells, less infectious particles are produced compared to BE/ANT-B13/17, suggesting either that the particles may not be efficiently released in the supernatant and remain more cell-associated or that the particles may efficiently attach but replicate at lower levels.

The same experiment was repeated in the A549 (Figure 3) cell line in which for the RSV-A isolates (Figure 3A), the RSV A2 shows the highest percentage of infected cells, followed closely by the BE/ANT-A11/17, reaching only slightly lower percentages of infected cells than in the HEp-2 cells after 72 h. This strain also produced the highest amounts of infectious virus in A549 cells (Figure 3C). In HEp-2 cells both the RSV-B subtype isolates BE/ANT-B13/17 and BE/ANT-B20/17 reach a higher percentage of infected cells than the RSV B1, whereas results of A549 replication kinetics suggest that the BE/ANT-B13/17 and BE/ANT-B2/17 strains reach similar infection rates (Figure 3B). The BE/ANT-B20/17 reached about 50% infection after 48 h but the infection then seemed to flatten out towards 72 h, resulting in a significant difference with infection rates of the RSV B1. Interestingly, the isolate BE/ANT-B2/17, which did not efficiently infect HEp-2 cells now reached a near 100% infected number in A549 cells in 72 h. Unsurprisingly, the BE/ANT-B15/17 achieved again the lowest number of infected cells and levels of virus production in A549 cells (Figure 3D).

As the BEAS-2B cell line is also a highly permissive cell line for RSV infection and widely used as well, we assessed viral growth and production kinetics in this cell line (Figure 4). For all RSV-A clinical isolates, no major differences were observed after 48 h and 72 h of infection in percentage of infected cells (Figure 4A). After 72h of infection, the amount of infectious virus released by the cells was the highest for RSV A2 and clinical isolate BE/ANT-A11/17. Larger differences were observed between the clinical isolates of the RSV-B subtype (Figure 4B). BE/ANT-B13/17 reached percentages and infectious virus production that were comparable to RSV B1 (Figure 4B,D). Isolates BE/ANT-B2/17 and BE/ANT-B15/17 had the lowest infection rates and infectious virus production in both this cell line as well as in the HEp-2 cells (Figure 4B,D).

Overall, clinical isolate BE/ANT-A11/17 replicated very efficiently in all cell lines, and remarkably, achieving even higher infection rates in the HEp-2 cell line than the RSV A2. Additionally, two clinical isolates of the RSV-B subtype (BE/ANT-B20/17 and BE/ANT-B13/17) replicated very well in HEp-2 and A549 cell lines and quite well in BEAS-2B. Overall, differences in infection kinetics were observed also among the different clinical isolates.

### 3.4. Thermal Stability

Differences in the F protein are shown to be involved in thermal stability of viral particles [37]. Aliquots of each virus containing 1×10^5^ PFU/mL were incubated at three different temperatures: 37 °C (in vitro incubator temperature and core body temperature) (Figure 5A,B), 32 °C (upper airway temperature) (Figure 5C,D) and 4 °C (storage temperature) (Figure 5E,F) for 24 h, 48 h and 72 h. Aliquots were snap frozen in liquid nitrogen and used for conventional plaque assay to quantify the remaining infectious virus. For all RSV-A isolates and RSV A2, higher temperatures were associated with a faster decay of infectious virus. Curiously, BE/ANT-A11/17 maintained higher PFU at 4 °C than other RSV-A isolates although at the other temperatures, there was no difference. Additionally, BE/ANT-A18/17 was preserved slightly better at 4 °C, however at 72 h no viable virus was detected. RSV-B isolate BE/ANT-B20/17 retained higher titers for the duration of the experiment compared to other RSV-B isolates but its overall stability was less than the reference RSV B1. The only exception is at 32 °C, where its viral titers remained higher than RSV B1. Isolate BE/ANT-B15/17 seems to decay especially fast at any other temperature than 37 °C.

### 3.5. Cell to Cell Fusion

Syncytium formation has long been considered a typical characteristic of RSV infection in immortal cell lines, and it has been used as a measure of activity of the fusion protein [27]. HEp-2 cells were infected at a MOI of 0.05 and incubated for 48 h with an overlay of DMEM-10 containing 0.6% Avicel^®®^ to allow spreading of the infection to neighbouring cells only. Afterwards, cells were fixed, fluorescently stained and analysed with fluorescence microscopy. Mean syncytium size was determined by calculating the mean of the number of nuclei in the syncytia formed per sample, (Figure 6A,B) as well as mean syncytium frequency (Figure 6C,D) by counting the number of nuclei belonging to syncytia relative to the total number of nuclei of infected cells. Mean syncytium size of all RSV-A clinical isolates (Figure 6A) lies between four and seven nuclei per cell, with BE/ANT-A1/16, BE/ANT-A8/17 and BE/ANT-A10/17 having the largest syncytia. The smallest syncytia were produced by BE/ANT-A12/17. Mean syncytium frequencies lie between 16% and 21%, with the lowest frequency found for BE/ANT-A10/17, which suggested that it promotes the formation of larger syncytia rather than many small syncytia (Figure 6C). Clinical isolate BE/ANT-B20/17 formed significantly larger syncytia with a mean size of 13 compared to all clinical isolates (Figure 6B). Reference strain RSV B1 formed almost no syncytia, with the smallest size and lowest frequency of all viruses tested.

### 3.6. Plaque Reduction by Palivizumab

Viral neutralization by palivizumab was assessed with a conventional plaque reduction assay. Virus was incubated with a two-fold dilution series of palivizumab for 1h at 37 °C and then transferred to HEp-2 cells for 2h at 37 °C to allow infection of non-neutralized virus. Afterwards, the supernatant was replaced by DMEM-10 containing 0.6% Avicel^®®^ and incubated for three days until plaques were visible to the naked eye. Plaques were counted to determine the concentration of palivizumab that neutralizes the virus by 50%.

Figure 7 shows that RSV-A clinical isolates BE/ANT-A7/17 and BE/ANT-A21/17 are 50% neutralized at lower palivizumab concentrations than most of the other clinical isolates and RSV A2, resulting in better neutralization than the other isolates. Remarkably, RSV A2 and RSV B1 neutralization was significantly different, with RSV-B strains having a higher sensitivity to palivizumab than RSV-A strains. Overall no significant differences were observed between RSV-B clinical isolates and the reference RSV B1 for palivizumab neutralization.

### 3.7. Mucin Expression

RSV infection is hallmarked by an increase of mucus production and impaired mucociliary clearance. As MUC5AC and MUC5B are important players in the secreted airway mucins and MUC1 and MUC4 in the cell-tethered mucins [29], their mRNA expression levels upon RSV infection of A549 cells was tested. mRNA expression levels of the mucins were assessed by infecting A549 cells for 48 h with a MOI of 0.1, followed by qRT-PCR with primers for the different mucin-encoding genes. A549 cells were incubated with virus of each isolate for 2 h, after which the inoculum was removed and replaced with DMEM-10. Cells were incubated for 48 h, collected for lysis followed by an RNA extraction and qRT-PCR.

For all clinical isolates and controls, the relative expression of cell-tethered MUC1 (Figure 8A) is increased compared to the non-infected control. No significant differences can, however, be observed in between RSV isolates and controls.

Expression profiles of the cell-tethered MUC4 show a considerable relative increase compared to the negative control (Figure 8B). Infection of BE/ANT-A1/16 and BE/ANT-A11/17 resulted in the highest relative increases of MUC4 mRNA among all of the RSV-A clinical isolates, whereas BE/ANT-A7/17 and BE/ANT-A12/17 resulted in the lowest increase. For the RSV-B clinical isolates, significantly lower increases are observed when compared to the RSV-A clinical isolates, but an increase is still observed. Infection of isolates BE/ANT-B13/17 and BE/ANT-B20/17 resulted in the highest increase of MUC4 mRNA expression among the RSV-B isolates.

MUC5AC is mainly produced in the epithelial goblet cells and was previously reported to slightly decrease in A549 cells under the influence of an RSV-infection after 48h [38]. Here, expression of MUC5AC is significantly reduced upon infection with all clinical isolates and reference strains, however no significant differences can be observed between the clinical isolates (Figure 8C).

MUC5B is produced by surface secretory cells throughout the airways and submucosal glands. Our results show that MUC5B expression is downregulated as a result of RSV infection, with the strongest downregulation of RSV-A clinical isolates BE/ANT-A1/16, BE/ANT-A7/17, BE/ANT-A11/17 and BE/ANT-A12/17. Overall downregulation of MUC5B by the RSV-B clinical isolates is limited, with almost none in infections with BE/ANT-B15/17 (Figure 8D).

## 4. Discussion

We have isolated RSV from nasal samples from patients in the winter seasons of 2016–2017 and 2017–2018 and found that isolating infectious virus from nasal samples is most efficient from nasopharyngeal aspirates (50%) compared to nasopharyngeal swabs (27%). However, yield may also depend on timing of sampling compared to onset of symptoms and time between sampling and inoculation of cell culture. HEp-2 cells were used for isolation since they are permissive for most RSV strains and isolates described. One well-known exception is RSV B1, which infects HEp-2 cells, but viral titers remain low compared to other strains. However, infection of RSV B1 in Vero cells does yield a higher titer, which is also the case for the clinical isolate BE/ANT-B20/17 from this study. The clinical isolates were genotyped and characterized on their ability to grow and produce infectious virus in cell lines. Thermal stability was assessed at 37 °C (normal body temperature, incubation temperature), 32 °C (estimated nasal temperature) and 4 °C (storage temperature), fusion capacity and neutralization by palivizumab as specific features of the fusion protein and mucin mRNA expression of MUC1 and MUC4 as cell-tethered mucins, and MUC5AC and MUC5B as secreted mucins.

Even though all RSV-A and RSV-B clinical isolates belong to the same genotype in the phylogenetic trees (ON1 and BAIX, respectively), the results show differences between the different isolates. According to the phylogenetic trees, the clinical isolates can be divided into three groups for the RSV-A isolates and two groups for the RSV-B isolates. However, differences can even be observed between clinical isolates that belong to the same group within the genotype. For example BE/ANT-A11/17, which, in comparison with BE/ANT-A8/17 and other clinical isolates from the same season, infects a higher number of cells in viral replication kinetics experiments, produced a higher amount of infectious virus and retained its stability in thermal stability assays. It is only slightly less neutralized by palivizumab compared to other RSV-A clinical isolates that have been isolated from the same season. Secondly, BE/ANT-B2/17 and BE/ANT-B15/17 are generally poor infectors and also have a lower thermal stability than other RSV-B clinical isolates from the same season. Generally, differences can be observed between the different clinical isolates in viral replication kinetics, but all clinical isolates have the ability to infect all cell lines tested. These differences could be due to underlying genetic differences that could affect antiviral responses in the cells, or due to differences that could be accounted to changes in the way the virus replicates in the cells, such as the presence of defective interfering virus particles. Our results suggest that the thermal stability of the clinical isolates is in general lower than the reference strains. The reason for this is unknown but it may be that repeated infection of the reference strains has resulted in the selection of variants with replicative fitness advantage at these temperatures. Remarkably, RSV B1 remains rather stable at 4 °C compared to the RSV A2 and other clinical strains.

The F protein is indispensable for virus infection [39] and its activity can be linked to the formation of syncytia [36,40]. Mean syncytium size and mean syncytium frequency were determined after 48h of infection with a 0.6% Avicel^®®^ overlay to minimize free particle movement. The presence of Avicel^®®^ would promote the formation of syncytia rather than spread and the formation of new syncytia during the incubation period. Even with minimal free particle movement, different sizes of syncytia containing 2–180 nuclei per cell are formed by the clinical isolates. For example, BE/ANT-B20/17 clearly forms large syncytia without infected satellite cells, which suggests that the produced particles remain more cell associated, and virus spread in HEp-2 cells may be facilitated through syncytia formation rather than particle release. This could also be observed from the replication kinetics and infectious virus production: the number of infected cells rapidly increases, and about 80% of the culture is infected after 48h, whereas the amount of infectious virus in the supernatant is lower compared to most other clinical isolates. This suggests that the production of infectious virus particles is either limited or faulty in HEp-2 cells for this isolate. A full characterization of the production of defective interfering particles could provide an explanation.

We tested the neutralization of the clinical isolates by palivizumab, the monoclonal antibody currently used to passively immunize infants with risk factors of severe RSV-related respiratory infections. Curiously, all RSV-B isolates and RSV B1 are better neutralized than all RSV-A clinical isolates.

Increased mucus production in the airways is an important characteristic and defence mechanism of the body to protect the airways from respiratory infections [29,38]. It has been shown that expression profiles of mucin mRNA can change under the influence of the RSV infection [38]. Relative expression of MUC1 and MUC4 was increased in the infected cells compared to a non-infected control whereas relative expression of MUC5AC and MUC5B was decreased in the infected cells compared to the negative control.

We have observed differences in characteristics between viruses which were isolated from children in one RSV season, such as in vitro viral replication kinetics in cell lines, infectious virus production, thermal stability, fusion capacity and neutralization-sensitivity. Additionally, mucin mRNA expression was assessed and revealed generally minor differences between the different clinical isolates. Furthermore, a comparison of the clinical isolates to the reference strains RSV A2 and RSV B1 reveal overall no major differences. Clinical isolates can replicate either faster or slower in cell lines, generally have a lower thermal stability, and have differences in fusion capacity and mucin expression.

## Figures and Tables

**Figure 1 viruses-11-01031-f001:**
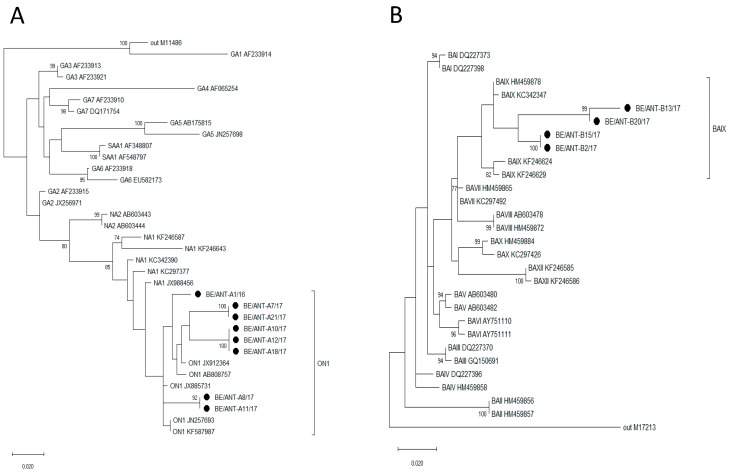
Phylogenetic trees for RSV-A and RSV-B clinical isolates. The phylogenetic trees were constructed with maximum-likelihood with 1000 bootstrap replicates using MEGA X software. The trees are based on a 342nt and 330nt fragment of the G protein of RSV-A (**A**) and RSV-B (**B**) strains respectively, consisting of the second hypervariable region. Nucleotide sequences of the clinical isolates (indicated with •) were compared to reference strains found on GenBank (indicated with genotype and accession number). The outgroups are represented by prototype strains M11486 for RSV-A and M17213 for RSV-B. Bootstrap values greater than 70% are indicated at the branch nodes and the scale bar represents the number of substitutions per site.

**Figure 2 viruses-11-01031-f002:**
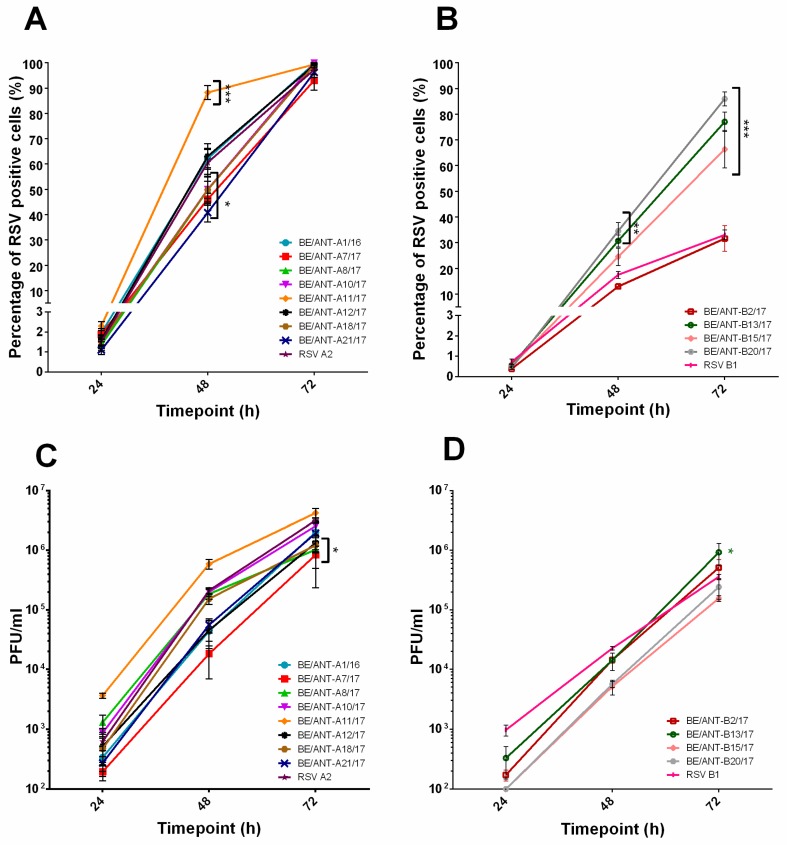
Replication kinetics and infectious virus production in HEp-2 cells. (**A**,**B**) HEp-2 cells were infected with clinical isolates and RSV reference strains A2 and B1. Cultures were fixed after 24 h, 48 h and 72 h, permeabilized and stained with polyclonal antibody (pAb) goat-anti-RSV antibody and AF488 donkey-anti-goat (IgG). Nuclei were visualized with DAPI and cultures were analysed with fluorescence microscopy. RSV positive cells were counted and calculated to the total number of nuclei to reach a percentage of RSV infected cells. (**A**) Replication kinetics of RSV-A clinical isolates and (**B**) Replication kinetics of RSV-B clinical isolates. (**C**,**D**) HEp-2 cells were infected with clinical isolates and RSV reference strains A2 and B1. After 24 h, 48 h and 72 h, supernatants were collected and used for quantification by conventional plaque assay. Data represents mean values ± SEM (*n* = 3), significant differences compared to the reference strains are indicated by **p* < 0.05; ****p* < 0.001 (two-way ANOVA).

**Figure 3 viruses-11-01031-f003:**
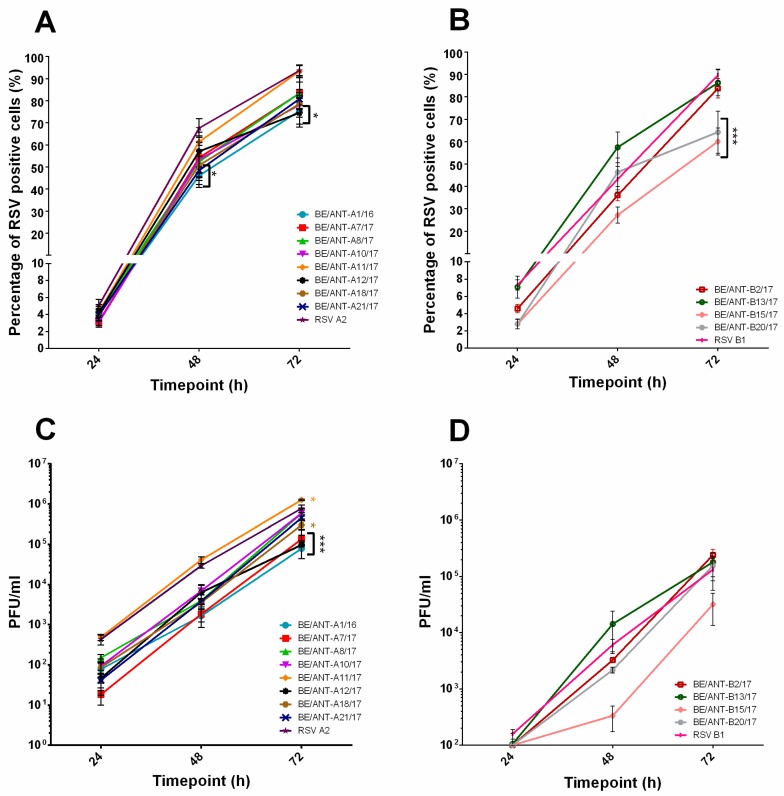
Replication kinetics and infectious virus production in A549 cells. (**A**,**B**) A549 cells were infected with clinical isolates and RSV reference strains A2 and B1. Cultures were fixed after 24 h, 48 h and 72 h, permeabilized and stained with pAb goat-anti-RSV antibody and AF488 donkey-anti-goat (IgG). Nuclei were visualized with DAPI and cultures were analysed with fluorescence microscopy. RSV positive cells were counted and calculated to the total number of nuclei to reach a percentage of RSV infected cells. (**A**) Replication kinetics of RSV-A clinical isolates and (**B**) Replication kinetics of RSV-B clinical isolates. (**C**,**D**) A549 cells were infected with clinical isolates and RSV reference strains A2 and B1. After 24 h, 48 h and 72 h, supernatants were collected and used for quantification by conventional plaque assay. Data represents mean values ± SEM (*n* = 3), significant differences compared to the reference strains are indicated by **p* < 0.05; ****p* < 0.001 (two-way ANOVA).

**Figure 4 viruses-11-01031-f004:**
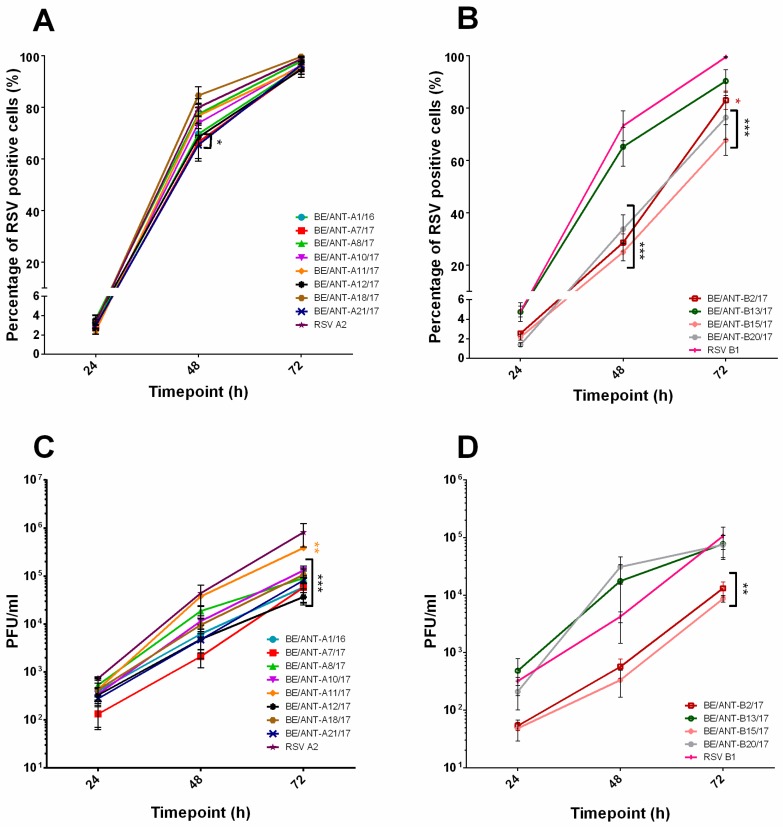
Replication kinetics and infectious virus production in BEAS-2B cells. (**A**,**B**) BEAS-2B cells were infected with clinical isolates and RSV reference strains A2 and B1. Cultures were fixed after 24 h, 48 h and 72 h, permeabilized and stained with pAb goat-anti-RSV antibody and AF488 donkey-anti-goat (IgG). Nuclei were visualized with DAPI and cultures were analysed with fluorescence microscopy. RSV positive cells were counted and calculated to the total number of nuclei to reach a percentage of RSV infected cells. (**A**) Replication kinetics of RSV-A clinical isolates and (**B**) Replication kinetics of RSV-B clinical isolates. (**C**,**D**) BEAS-2B cells were infected with clinical isolates and RSV reference strains A2 and B1. After 24 h, 48 h and 72 h, supernatants were collected and used for quantification by conventional plaque assay. Data represents mean values ± SEM (*n* = 3), significant differences compared to the reference strains are indicated by **p* < 0.05; ***p* < 0.01 ***; *p* < 0.001 (two-way ANOVA).

**Figure 5 viruses-11-01031-f005:**
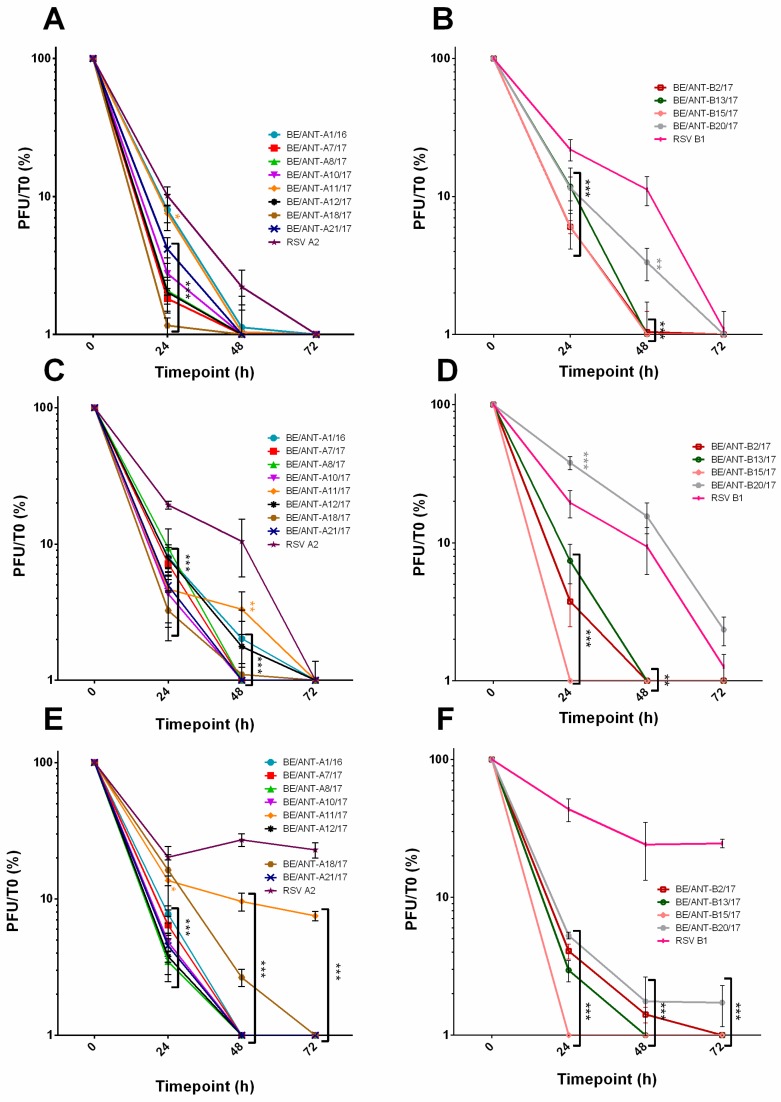
Thermal stability profiles at 37 °C, 32 °C and 4 °C. Clinical isolates, RSV A2 and RSV B1 were aliquoted and exposed to 37 °C (**A**,**B**), 32 °C (**C**,**D**) or 4°C (**E**,**F**). One aliquot of each was snap frozen at 0 h, 24 h, 48 h and 72 h. Aliquots were used for quantification by conventional plaque assay and calculated to the amount at 0 h. Data represents mean values ± SEM (*n* = 3), significant differences compared to the reference strains are indicated by **p* < 0.05;***p* < 0.01; ****p* < 0.001 (two-way ANOVA).

**Figure 6 viruses-11-01031-f006:**
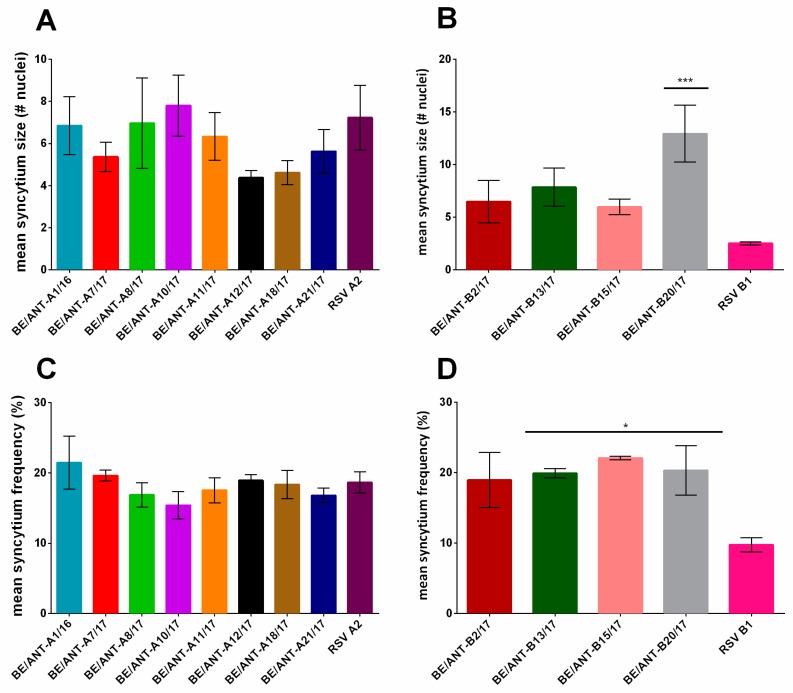
The capacity for syncytium formation of clinical isolates. HEp-2 cells were infected with clinical isolates and RSV reference strains A2 and B1 for 2 h, inoculum was replaced by DMEM-10 containing 0.6% Avicel^®®^ and incubated for 48 h at 37 °C. Afterwards, cells were fixed, permeabilized and stained with pAb goat-anti-RSV and AF488 donkey-anti-goat. Nuclei were visualized with DAPI and cultures were analysed with fluorescence microscopy. (**A**, **B**) Mean syncytium size was calculated by counting the number of nuclei in syncytia in three pictures taken at 10× magnification. (**C**,**D**) Mean syncytium frequency was calculated by dividing the number of syncytial cells by the total number of infected cells. Data represents mean values ± SEM (*n* = 3), significant differences compared to the reference strains are indicated by **p* < 0.05; ****p* < 0.001 (one-way ANOVA).

**Figure 7 viruses-11-01031-f007:**
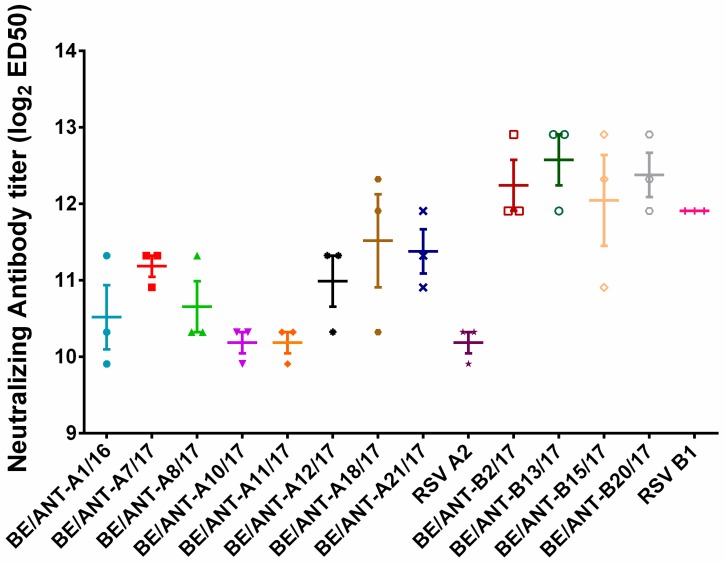
Plaque reduction of the clinical isolates with palivizumab. HEp-2 cells were infected for 2h with clinical isolates and reference strains that were pre-incubated for 1h with a palivizumab dilution series. Inoculum was replaced with DMEM-10 containing 0.6% Avicel^®®^ and incubated for three days at 37°C. Afterwards, the cells were fixed, stained with palivizumab as primary antibody and goat-anti-human conjugated with HRP, plaques were visualized with chloronapthol. Individual values are plotted as 2log ED50, data represents mean values ± SEM (*n* = 3).

**Figure 8 viruses-11-01031-f008:**
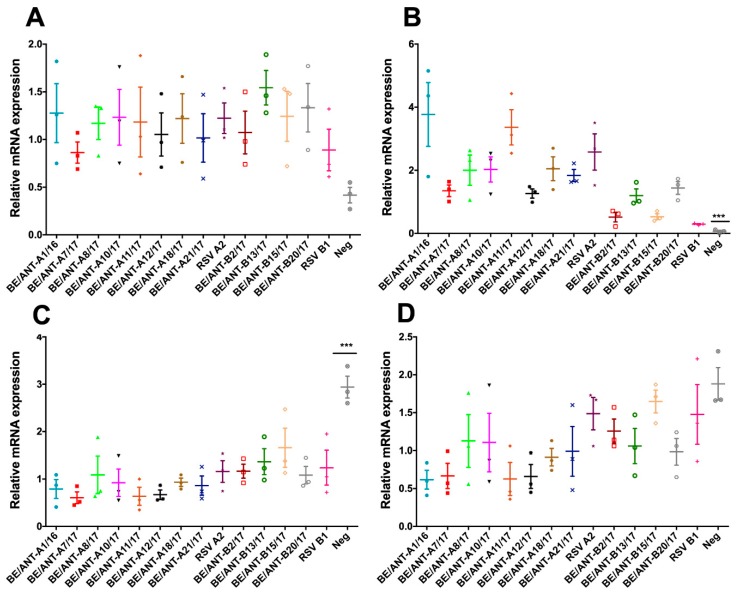
mRNA levels of mucins 1, 4, 5AC and 5B in infected A549 cells. A549 cells were infected with a MOI of 0.1 of clinical isolates and reference strains for 2 h at 37 °C. Inoculum was replaced with DMEM-10 and cells were incubated for 48h at 37 °C. Afterwards, cells were lysed, total RNA was extracted and the expression of MUC1 (**A**), MUC4 (**B**), MUC5AC (**C**) and MUC5B (**D**) was determined by qRT-PCR. Data represents mean values ± SEM (*n* = 3), statistically significant differences compared to the reference strains are indicated with ****p* < 0.001 (one-way ANOVA).

**Table 1 viruses-11-01031-t001:** Overview of clinical isolates and viruses used in experiments, with subtyping results and cell type used for propagation.

NAME:	SUBTYPE:	GROWN ON:	VIRUS ISOLATED FROM:	EXPERIMENTS PERFORMED WITH:
**BE/ANT-A1/16**	RSV-A	HEp-2	Aspirate	Aspirate
**BE/ANT-B2/17**	RSV-B	HEp-2	Aspirate/Swab	Aspirate
**BE/ANT-A7/17**	RSV-A	HEp-2	Aspirate/Swab	Aspirate
**BE/ANT-A8/17**	RSV-A	HEp-2	Aspirate	Aspirate
**BE/ANT-A10/17**	RSV-A	HEp-2	Swab	Swab
**BE/ANT-A11/17**	RSV-A	HEp-2	Aspirate	Aspirate
**BE/ANT-A12/17**	RSV-A	HEp-2	Aspirate/Swab	Aspirate
**BE/ANT-B13/17**	RSV-B	HEp-2	Aspirate	Aspirate
**BE/ANT-B15/17**	RSV-B	HEp-2	Aspirate/Swab	Aspirate
**BE/ANT-A18/17**	RSV-A	HEp-2	Aspirate	Aspirate
**BE/ANT-B20/17**	RSV-B	Vero	Aspirate	Aspirate
**BE/ANT-A21/17**	RSV-A	HEp-2	Aspirate/Swab	Aspirate
**RSV A2**	RSV-A	HEp-2	/	
**RSV B1**	RSV-B	Vero	/

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
