# Peer review of "Isolation and Characterization of Clinical RSV Isolates in Belgium during the Winters of 2016–2018"

_viruses, 2019, doi:10.3390/v11111031_

Round 1

Reviewer 1 Report

This work aims at addressing whether RSV A and B reference strains used for the majority of laboratory experiments are representative of the currently circulating viruses. The authors compared 8 RSV A isolates and 4 RSV B isolates to the lab-adapted RSV A2 and RSV B1 strains. They used multiple cell lines to assess viral growth kinetics, infectious virus production, thermal stability, syncytia formation, neutralization by palivizumab and mucin mRNA expression. The authors conclude that the reference strains are significantly distinct to currently circulating viruses.

Although highly relevant, the work as presented is majorly flawed and most of the conclusions raised are not supported.

Virus directly isolated from patients cannot be compared for virus kinetics and production, or effect on host cells unless thoroughly purified from other potential contaminants, including other pathogens (common RSV contaminants include other respiratory viruses such as adenovirus and parainfluenza virus), and defective viral particles that are preferentially amplified during virus expansion. These contaminants will affect the parameters measured here, and therefore without clear evidence of their absence data are not conclusive. Most experiments show no statistically significant differences. The experiments that do show variation show that the reference strains fall right in the middle with some isolates showing higher levels and others showing lower levels. If anything, the most interesting comparison is RSV A to RSV B characteristics, which are very different. Line 242-243: Do the authors mean that the one patient in December 2016 had RSV-A and 11 patients (out of 24) between October and January 2017-2018 had RSV-A. The 2016-2017 and 2017-2018 periods overlap, which makes this sentence unclear. Table 1: It is interesting that 8 out 12 (67%) RSV-A samples and only 4 out of 11 (36%) RSV-B samples were recovered in Hep-2 cells. Furthermore, 1 RSV-B sample had to be grown on Vero cells. Would using Vero cells for RSV B recovery would have been more productive? What explanation could there be for recovering less RSV-B samples? Figure 2: If 0% of cells were infected at 24hpi how was there already infectious particles being produced at 24hpi? Also an earlier time point would have been useful to confirm that equal amount of virus (in addition to MOI) was used. Microscopy pictures would be also useful here especially for the significant results. Line 287-288: Other option is that they attach more efficiently and as a consequence infect most cells but have lower replication levels. In addition, to the production of infectious particles it would be interesting to know the replication levels. The authors could for example track RSV replication levels by qPCR, which would also be more sensitive and allow earlier time points such a 2h, 6h and 12h. Or the replication levels could be extracted from the images the authors have taken. Line 318: More specifics about antibody (against what RSV protein) are needed. Figure 2, 3 and 4: The ability of viruses to replicate depends a lot on the innate immune response to the virus. It would be very interesting to see the expression by qPCR of IFN alpha, beta and lambda and ISGs in the different cell lines in response to these viruses. Different immunostimulatory responses in different cell lines could explain the differences observed. Figure 2, 3 and 4: It would be interesting to see similar results done in primary cells. It would be interesting to know how the virus sequence changed by being passaged in Hep-2 or Vero cells. How similar are the stock compared to original sample?

Author Response

Open Review (Reviewer 1)

Comments and Suggestions for Authors

This work aims at addressing whether RSV A and B reference strains used for the majority of laboratory experiments are representative of the currently circulating viruses. The authors compared 8 RSV A isolates and 4 RSV B isolates to the lab-adapted RSV A2 and RSV B1 strains. They used multiple cell lines to assess viral growth kinetics, infectious virus production, thermal stability, syncytia formation, neutralization by palivizumab and mucin mRNA expression. The authors conclude that the reference strains are significantly distinct to currently circulating viruses.

Although highly relevant, the work as presented is majorly flawed and most of the conclusions raised are not supported.

Virus directly isolated from patients cannot be compared for virus kinetics and production, or effect on host cells unless thoroughly purified from other potential contaminants, including other pathogens (common RSV contaminants include other respiratory viruses such as adenovirus and parainfluenza virus), and defective viral particles that are preferentially amplified during virus expansion. These contaminants will affect the parameters measured here, and therefore without clear evidence of their absence data are not conclusive.

The virus was isolated from patients and purified through cell culture. The virus cultures that were used for the experiments described in this paper were assessed afterwards by qPCR in which the following possible contaminants were tested and found absent from the viral supernatant: hMPV, Rhinovirus 1/3, PIV1/2/3/4, adenovirus, Coronavirus 229E/NL63/OC43, Paraechovirus, Enterovirus 68 and Bocavirus. Several virus cultures were tested on mycoplasma presence and were found negative, and no signs of other bacterial or fungal presence was observed when the cultures were test-grown in absence of antibiotics.This was added to the text.

Most experiments show no statistically significant differences. The experiments that do show variation show that the reference strains fall right in the middle with some isolates showing higher levels and others showing lower levels. If anything, the most interesting comparison is RSV A to RSV B characteristics, which are very different.

We agree with the reviewer on the absence of statistical differences. However, the scope of this paper was to isolate and compare the resulting viruses with the prototype strains. In some characteristics, significant differences can be observed, whereas in others there cannot, which is an interesting finding in our opinion. The significant differences throughout the different experiments were found to be from different viruses. This indicates and is an important conclusion in our manuscript, that there is variation between the circulating viruses and that they differ in specific characteristics.

Line 242-243: Do the authors mean that the one patient in December 2016 had RSV-A and 11 patients (out of 24) between October and January 2017-2018 had RSV-A. The 2016-2017 and 2017-2018 periods overlap, which makes this sentence unclear.

We agree with the reviewer that this sentence is not fully clear, and have adapted the sentence in the manuscript: “Nasal swabs and nasopharyngeal aspirates were obtained from one patient in December of 2016 and from 24 patients between October and January 2017-2018. RSV-A was detected in one sample of December 2016 and in 11 samples collected between October 2017 and January 2018. RSV-B was detected in 11 samples collected between October 2017 and January 2018.”

Table 1: It is interesting that 8 out 12 (67%) RSV-A samples and only 4 out of 11 (36%) RSV-B samples were recovered in Hep-2 cells. Furthermore, 1 RSV-B sample had to be grown on Vero cells. Would using Vero cells for RSV B recovery would have been more productive? What explanation could there be for recovering less RSV-B samples?

We would like to thank the reviewer for this insightful comment. Many factors could play a role in the lower recovery of RSV-B samples in comparison to RSV-A samples. Firstly, at the time of nasal extraction, we did not know if the sample was negative or positive, and did not know if the sample was RSV-A or RSV-B. We have chosen to recover on the HEp-2 cell line as it has the highest affinity for growing RSV positive samples. Secondly, we have tested if we could still recover RSV-B samples on Vero cells after the first freeze-thaw cycle which did not result in any virus recovery. Furthermore, more samples have been collected in the latest winter season between October 2018 and January 2019, in which the percentage of RSV-B recovered samples was much higher than the RSV-A recovered samples. We do not foresee an exact explanation for the lower recovery of RSV-B samples, but theorize that time of extraction would be the most important indicator.

Figure 2: If 0% of cells were infected at 24hpi how was there already infectious particles being produced at 24hpi? Also an earlier time point would have been useful to confirm that equal amount of virus (in addition to MOI) was used. Microscopy pictures would be also useful here especially for the significant results.

We agree with the reviewer that it may seem that the percentage of infected cells at 24hpi is 0%, however the values lie around 1% which is not visible in the figure. We have adapted the figures.

Line 287-288: Other option is that they attach more efficiently and as a consequence infect most cells but have lower replication levels. In addition, to the production of infectious particles it would be interesting to know the replication levels. The authors could for example track RSV replication levels by qPCR, which would also be more sensitive and allow earlier time points such a 2h, 6h and 12h. Or the replication levels could be extracted from the images the authors have taken.

We thank the reviewer for this suggestion. Indeed this is one of the different possible explanations for our observations and this has been added to the text. More detailed analysis on the cause of these differences will be done in a follow-up study and the suggestion will certainly be taken into account.

Line 318: More specifics about antibody (against what RSV protein) are needed.

The polyclonal antibody used in these experiments is a commercial antibody from Virostat. The polyclonal antibody is reported as “immunogen: Human RSV isolate (confirmed); specificity: All viral antigens”. This has been added to the methods section

Figure 2, 3 and 4: The ability of viruses to replicate depends a lot on the innate immune response to the virus. It would be very interesting to see the expression by qPCR of IFN alpha, beta and lambda and ISGs in the different cell lines in response to these viruses. Different immunostimulatory responses in different cell lines could explain the differences observed.

We thank the reviewer for this suggestion and agree that further analysis of this aspect is interesting. Further investigations to gain a better understanding and explanation of the observed differences is, as mentioned in comment (6) above, the scope of a follow-up study and this suggestion will certainly be taken into account

Figure 2, 3 and 4: It would be interesting to see similar results done in primary cells.

We fully agree with the reviewer that such experiments would be very interesting and this is something that is planned for future experiments with a selection of primary isolates.

It would be interesting to know how the virus sequence changed by being passaged in Hep-2 or Vero cells. How similar are the stock compared to original sample?

We were also interested in this matter and have compared some of the sequences of the G-gene of the original sample with the ones derived from the passage 3 virus and did not find significant base changes in these sequences, indicating that the stocks are still very similar to the original sample

Reviewer 2 Report

The authors collected different RSV samples from two recent seasons and investigated several parameters of these viruses. They conclude that there is some variation between clinical RSV strains and that there is a significant difference from laboratory strains. These are relevant observations for RSV research but it remains unclear if there might be a clinical impact.

The authors may want to make a statement on the correlation between the observed differences and the disease outcome. Is there information on the patient status and the different strains? Might it be that they collected more virulent strains as less virulent strains do not cause hospitalisation (and therefore no sample collection)? Or can there be no conclusions drawn on the clinical impact of the differences observed?

line 28: I'm not sure if there is a standard in defining seasons but I believe "winters 2016-2017 and 2017-2018" is more clear.

line32: We observe that viruses isolated in one RSV season show major differences on the tested assays.

line40: "... in infants, children and adult patients with ... disease and it is recognized ..."

In the methods the amount of cells per well of a plate is often indicated as cells/ml without the volume. Better to indicate as cells/well

line152: "final concentration of 30 pmol" is this pM or pmol/µL?

line170: remove "followed by inoculation"

line243: remove "also"

Include extra colums in table 1: "swap or aspirate" and "2016-2017 or 2017-2018"

in the text the expression "replication kinetics" is used while in the figures it is referred to as "growth kinetics", better to be consistent

the units of EC50 on the Y-axis of fig7 is not indicated. Is this mg/mL or µg/mL or ...?

line401 "that neutralizes the virus by 50%"

line464 "all clinical isolates have the same genotype" perhaps better "all RSV-A and RSV-B clinical isolates have the same genotype (ON1 and BA respectively)."

line468: the sentence that start with "Whereas ..." is difficult to read. In the previous line you indicate there are differences between the isolates of the same group. Is this new sentence than an example? But should you then not compare BE/ANT-A11/17 with other RSV-A isolates?

line475: The authors suggest that using reference strains for a long time would improve their thermal stability. I suppose they mean this is by Darwinian selection. But what is the selection pressure? Is it perhaps the repeated freeze/thaw cycles these reference strains are going through that does the selection? Or is it the temperature difference between a labincubator (37°C) and the tissue where the virus is isolated from (35°C perhaps?) that drives the selection? Or ...?

line495 "This would suggest that ..." Perhaps the authors can check if there is already literature on this topic. I assume others have already explored this.

line502: 'This increase' should be 'This difference'. The authors write here that A549 cannot produce MUC5AC and MUC5B while in Fig8 the relative expression levels of uninfected cells is higher for MUC5AC and MUC5B than the other mucins. Perhaps the authors should simply state that the observations on MUC5AC and MUC5B are rather irrelevant because A549 is an irrelevant cell line to study this. For the same reason it could be left out of Fig8.

Author Response

Open Review (Reviewer 2)

Comments and Suggestions for Authors

The authors collected different RSV samples from two recent seasons and investigated several parameters of these viruses. They conclude that there is some variation between clinical RSV strains and that there is a significant difference from laboratory strains. These are relevant observations for RSV research but it remains unclear if there might be a clinical impact.

The authors may want to make a statement on the correlation between the observed differences and the disease outcome. Is there information on the patient status and the different strains? Might it be that they collected more virulent strains as less virulent strains do not cause hospitalisation (and therefore no sample collection)? Or can there be no conclusions drawn on the clinical impact of the differences observed?

At this time, with the data obtained from the patients and the current number of viruses, we are unable to make a statement considering the observed differences and the clinical impact. We however agree with the reviewer that collecting more strains is interesting to be able to correlate virulent strains with hospitalizations or disease severity and such studies are ongoing. Nevertheless, in our experiments, we can clearly see differences on the conditions tested, even with the hospitalization criterium, which can indicate that similar differences would be observed if nasal secretions were taken from non-hospitalized patients.

line 28: I'm not sure if there is a standard in defining seasons but I believe "winters 2016-2017 and 2017-2018" is more clear.

We agree and have changed this in the text.

line32: We observe that viruses isolated in one RSV season show major differences on the tested assays.

We would like to thank the reviewer for his attention to detail and have changed this in the text.

line40: "... in infants, children and adult patients with ... disease and it is recognized ..."

We would like to thank the reviewer for his attention to detail and have changed this in the text.

In the methods the amount of cells per well of a plate is often indicated as cells/ml without the volume. Better to indicate as cells/well

This is correct and was changed in the manuscript.

line152: "final concentration of 30 pmol" is this pM or pmol/µL?

We would like to thank the reviewer for his attention to detail. The final amount is 30pmol.

line170: remove "followed by inoculation"

This has been removed in the manuscript.

line243: remove "also"

This has been removed in the manuscript.

Include extra colums in table 1: "swap or aspirate" and "2016-2017 or 2017-2018"

thank you for this suggestion, we have changed this in the manuscript.

in the text the expression "replication kinetics" is used while in the figures it is referred to as "growth kinetics", better to be consistent

We have modified the manuscript to be consistent, thank you.

the units of EC50 on the Y-axis of fig7 is not indicated. Is this mg/mL or µg/mL or ...?

We thank the reviewer for this comment, instead of EC50 in figure 7, it should be ED50, which is defined as the reciprocal of the highest antibody dilution producing 50% reduction in plaques (ED50), relative to virus-control wells without antibody. The method section is modified to better explain and the figure was adapted.

line401 "that neutralizes the virus by 50%"

This was corrected in the manuscript.

line464 "all clinical isolates have the same genotype" perhaps better "all RSV-A and RSV-B clinical isolates have the same genotype (ON1 and BA respectively)."

Thank you, we have changed this in the manuscript.

line468: the sentence that start with "Whereas ..." is difficult to read. In the previous line you indicate there are differences between the isolates of the same group. Is this new sentence than an example? But should you then not compare BE/ANT-A11/17 with other RSV-A isolates?

We agree with the reviewer that it was a difficult sentence and have adapted the section as follows:

For example the isolate BE/ANT-A11/17, which, in comparison with BE/ANT-A8/17 and other clinical isolates from the same season, infects a higher number of cells in viral replication kinetics experiments, produced a higher amount of infectious virus and retained its stability in thermal stability assays. It is only slightly less neutralized by palivizumab compared to other RSV-A clinical isolates that have been isolated from the same season.

line475: The authors suggest that using reference strains for a long time would improve their thermal stability. I suppose they mean this is by Darwinian selection. But what is the selection pressure? Is it perhaps the repeated freeze/thaw cycles these reference strains are going through that does the selection? Or is it the temperature difference between a labincubator (37°C) and the tissue where the virus is isolated from (35°C perhaps?) that drives the selection? Or ...?

We agree with the reviewer that this statement is too strong. We meant to say that we think that selection of the viruses over time may have resulted in variants with increased replicative fitness in the used cultures at the used temperatures and during the freeze-thaw cycles. We have changed this in the manuscript.

line495 "This would suggest that ..." Perhaps the authors can check if there is already literature on this topic. I assume others have already explored this.

We have revisited the literature on this topic, but are unable to find a statement on this. Nevertheless, because of suggestions of other reviewers, and because of the limited number of virus isolates tested, this sentence was removed from the manuscript

line502: 'This increase' should be 'This difference'. The authors write here that A549 cannot produce MUC5AC and MUC5B while in Fig8 the relative expression levels of uninfected cells is higher for MUC5AC and MUC5B than the other mucins. Perhaps the authors should simply state that the observations on MUC5AC and MUC5B are rather irrelevant because A549 is an irrelevant cell line to study this. For the same reason it could be left out of Fig8.

We would like to thank the reviewer to draw our attention to this. We revisited the literature and have decided to delete the comment on the production of mucins by A549 as it is a widely used cell line to study the production of mucins in the context of virus infection, but also outside the context of virus infections.

Reviewer 3 Report

In the present manuscript, Van der Gucht and coll. perform an interesting descriptive study that compares patient-derived RSV isolates from the A and B subtype with the two prototype strains that have been used for the last 50-60 years. Overall, this is a scientifically and technically well-conducted and well-written study. However, I believe that certain major points need to be addressed before considering the manuscript for publication.

Major concerns:  

1) If I have well understood from section 2.2 (lines 106-127), the so-called “passage 0” viruses have been already passaged 3 times beforehand (twice in 96-w and once in T25), which means that “passage 3” viruses are in fact passage 6. If this is the case, besides the need to correct this point, this raises the question of what the authors consider “low passage clinical isolates” (line 513) and the potential implications for the comparison with prototype strains. Please explain.

2) When showing data from experiments with less than 10 replicates, it is desirable to represent in the graph individual points with the mean/median and SD/SEM (as in Figure 7), since they are more informative than bars and SD/SEM in terms on reproducibility and dispersion among replicates. I therefore suggest re-formatting Figures 6 and 8 to match Figure 7.

3) Figure 8: it is not clear what the “relative mRNA expression” means. Normally, mRNA expression data for each MUC gene should be initially normalized by the GAPDH and/or B-actin and then expressed in relation to the normalized value obtained for the negative control (arbitrarily set at 1 in the graph). In addition, how was the negative control produced?  

4) What is the rationale behind the statement (lines 475-477) “Thermal stability of the clinical isolates is in general lower than the reference strains. This may not be surprising, as reference strains have been used for in vitro experiments for a long time and have in that process probably been selected on variants with increased thermal stability.”? I do not see the reason to explain the mild higher stability (observed mainly at 4 °C) as a result of serial passaging of prototype strains. Please explain.

5) Data on differential neutralization with palivizumab is quite interesting. However, a statement such as (lines 495-496) “This would suggest that palivizumab has a higher affinity for the F proteins expressed from RSV-B than that of RSV-A.” would be better supported by sequencing data on the different F genes, which should not be difficult to produce. Besides the importance of sequencing the genes coding for the major viral antigen for the comparative analysis proposed in this study, this could also highlight (or not) specific mutations affecting the affinity of palivizumab for the antigenic site II, as reported by J Bates and coll. (PMID 24725940, DOI: 10.1016/j.virol.2014.02.010).

Minor concerns and typo:

Line 71: Replace “21” by “Twenty-one”.

Lines 184 and 350: Please correct “1x105”.

Lines 233, 235, 237, etc and Figure legends: The “±” symbols are missing.

Lines 179, 210, 273 and 275: Replace “an MOI” by “a MOI”.

Figure 1: for clarity purposes, please indicate genotypes (ON1 and BA) in the figure.

Figures 2, 3 and 4: asterisks (*) in most of the panels are difficult to assign to the right dataset, for which it could be useful to color-code them to match the groups. Other options to improve clarity are also welcomed.

Line 298: Replace “efficiently infected” by “efficiently infect”.

Lines 298-299: Replace “near 100% infected number of cells” by “near 100% infection in A549 cells”.

Line 314: Replace “in between” by “among”.

Line 464: Replace “have the” by “belong to”.

Line 484: Is “2 to 180” correct?

Author Response

Reviewer 3

Comments and Suggestions for Authors

In the present manuscript, Van der Gucht and coll. perform an interesting descriptive study that compares patient-derived RSV isolates from the A and B subtype with the two prototype strains that have been used for the last 50-60 years. Overall, this is a scientifically and technically well-conducted and well-written study. However, I believe that certain major points need to be addressed before considering the manuscript for publication.

 Major concerns:  

If I have well understood from section 2.2 (lines 106-127), the so-called “passage 0” viruses have been already passaged 3 times beforehand (twice in 96-w and once in T25), which means that “passage 3” viruses are in fact passage 6. If this is the case, besides the need to correct this point, this raises the question of what the authors consider “low passage clinical isolates” (line 513) and the potential implications for the comparison with prototype strains. Please explain.

We understand that the passage numbers may be cause confusion. We considered the isolation procedure, which consists of taking the highest dilution of virus that still infects the cells, without freeze-thaw cycle as described in the materials and methods section, being different than a passage as a classic multiplication process, producing high amounts of virus, followed by a freeze-thaw cycle.

When showing data from experiments with less than 10 replicates, it is desirable to represent in the graph individual points with the mean/median and SD/SEM (as in Figure 7), since they are more informative than bars and SD/SEM in terms on reproducibility and dispersion among replicates. I therefore suggest re-formatting Figures 6 and 8 to match Figure 7.

We agree with the reviewer that individual points may be more informative, however we feel that for Figure 6 and 8, which contain many more data points, the easy comparability between the viruses and the overview will be lost when using the individual points with so many viruses. By using bar graphs, we can show the overall picture based on the average as bar height, the variation in the error bars and the presence and absence statistical significance.

Figure 8: it is not clear what the “relative mRNA expression” means. Normally, mRNA expression data for each MUC gene should be initially normalized by the GAPDH and/or B-actin and then expressed in relation to the normalized value obtained for the negative control (arbitrarily set at 1 in the graph). In addition, how was the negative control produced?  

The relative mRNA expression values were indeed normalized by GAPDH and B-actin, however instead of setting the value of the negative control at 1 in the graph, the reference genes were set at 1 and the relative expression was calculated as such. The negative control represents an A549 culture that was present in each experiment, following the same experimental steps as the other cultures except that there was no virus present in the “inoculum”.

What is the rationale behind the statement (lines 475-477) “Thermal stability of the clinical isolates is in general lower than the reference strains. This may not be surprising, as reference strains have been used for in vitro experiments for a long time and have in that process probably been selected on variants with increased thermal stability.”? I do not see the reason to explain the mild higher stability (observed mainly at 4 °C) as a result of serial passaging of prototype strains. Please explain.

We agree with the reviewer that this statement is too strong. We meant to say that we think that long term replication of the viruses may have resulted in variants with increased replicative fitness in the used cultures at the used temperatures and during the freeze-thaw cycles. We have changed this in the text.

Data on differential neutralization with palivizumab is quite interesting. However, a statement such as (lines 495-496) “This would suggest that palivizumab has a higher affinity for the F proteins expressed from RSV-B than that of RSV-A.” would be better supported by sequencing data on the different F genes, which should not be difficult to produce. Besides the importance of sequencing the genes coding for the major viral antigen for the comparative analysis proposed in this study, this could also highlight (or not) specific mutations affecting the affinity of palivizumab for the antigenic site II, as reported by J Bates and coll. (PMID 24725940, DOI: 10.1016/j.virol.2014.02.010).

We agree with the reviewer that this statement would be better supported by sequencing data; As mentioned above, we plan a follow-up study to better understand the mechanism behind the observed differences is planned and this will also include the sequencing of the viruses. Therefore, we have removed the sentence “This would suggest that palivizumab has a higher affinity for the F proteins expressed from RSV-B than that of RSV-A.” from the manuscript.

 We thank the reviewer for his attention to detail. All the following concerns have been taken care of.

Minor concerns and typo:

Line 71: Replace “21” by “Twenty-one”. Has been changed.

Lines 184 and 350: Please correct “1x105”. Has been changed.

Lines 233, 235, 237, etc and Figure legends: The “±” symbols are missing. Has been changed.

Lines 179, 210, 273 and 275: Replace “an MOI” by “a MOI”. Has been changed.

Figure 1: for clarity purposes, please indicate genotypes (ON1 and BA) in the figure.

We have adapted figure 1 to indicate ON1 and BAIX more clearly.

Figures 2, 3 and 4: asterisks (*) in most of the panels are difficult to assign to the right dataset, for which it could be useful to color-code them to match the groups. Other options to improve clarity are also welcomed.

We agree with the reviewer that it is unclear have slightly changed the graphs by color-coding some of the asterisks and using a different way to show statistical significance to clarify the figures.

Line 298: Replace “efficiently infected” by “efficiently infect”. Has been changed.

Lines 298-299: Replace “near 100% infected number of cells” by “near 100% infection in A549 cells”. Has been changed.

Line 314: Replace “in between” by “among”. Has been changed.

Line 464: Replace “have the” by “belong to”. Has been changed.

Line 484: Is “2 to 180” correct?

In our experiments evaluating the size of the syncytia, we observed syncytia of 2 nuclei, and up to 180 nuclei, seemingly in one cell.

Reviewer 4 Report

This manuscript describes a study of clinical isolates of respiratory syncytial virus. The study examines replication kinetics in several cell lines, thermal stability, syncytia formation, sensitivity to palivizumab, and mucin production, giving a fairly complete picture of the differences between RSV clinical isolates and the currently used lab strains. The manuscript is well-written and I believe will be of interest to the virology community. I recommend the manuscript be accepted after minor revisions.

Specific comments:

The discussion of the manuscript could be improved. While it nicely summarizes the results of the study, there is no consideration of what these results might mean for the clinical course of the illness caused by these different strains. In particular, the study found differences in sensitivity to palivizumab --- what are the implications for the continued use of this treatment? I'm wondering why the study only examined the first 72 h for replication kinetics. While most strains had infected the majority of cells at that point, some strains, particularly the RSV B strains showed much slower growth kinetics. The picture would be more complete had the study gone for one more day. Why do the authors use SEM as an estimate of error for some measurements, but SD for others? The plus/minus symbol is missing in the captions and in the methods section. line 39, Saying RSV is "the most important" viral respiratory pathogen is pretty strong. Influenza is also pretty serious in these groups. Line 76, "have been" should be "were" Line 90, dr should be Dr Line 269, should read "scale bar". Line 402, I'm not sure why Figure 7 is in bold.

Author Response

Reviewer 4

Comments and Suggestions for Authors

This manuscript describes a study of clinical isolates of respiratory syncytial virus. The study examines replication kinetics in several cell lines, thermal stability, syncytia formation, sensitivity to palivizumab, and mucin production, giving a fairly complete picture of the differences between RSV clinical isolates and the currently used lab strains. The manuscript is well-written and I believe will be of interest to the virology community. I recommend the manuscript be accepted after minor revisions.

Specific comments:

The discussion of the manuscript could be improved. While it nicely summarizes the results of the study, there is no consideration of what these results might mean for the clinical course of the illness caused by these different strains.

We agree with the reviewer that it would be very interesting, however we feel that we are unable to make a conclusion on the clinical course of the illness without including more RSV isolates and more data considering the innate immune response, which was outside the scope of this article. Isolation of RSV from additional patients and studies on the innate immune response differences are ongoing

In particular, the study found differences in sensitivity to palivizumab --- what are the implications for the continued use of this treatment?

Currently we are not exactly sure what the implications may be. We indeed see differences, yet these seem to rather limited. In the case of continued immune pressure from Palivizumab this may result in a more resistant phenotype as has been described in the past.

I'm wondering why the study only examined the first 72 h for replication kinetics. While most strains had infected the majority of cells at that point, some strains, particularly the RSV B strains showed much slower growth kinetics. The picture would be more complete had the study gone for one more day.

We thank the reviewer for his insightful comment. It is indeed true that for the RSV-B isolates, an additional experimental day would lead to higher percentages. The experiments however were performed evaluating the RSV-A clinical isolates, that at 96h post infection would have resulted in a disrupted culture that we would have been unable to stain and evaluate by fluorescence microscopy. Therefore, we chose to limit all experiments to 72h as this was the latest timepoint that allowed a fair comparison between all isolates

Why do the authors use SEM as an estimate of error for some measurements, but SD for others?

We thank the reviewer for noticing this error and have adapted all figures for consequent estimations of error.

The plus/minus symbol is missing in the captions and in the methods section.

This was an editing mistake and has been corrected.

line 39, Saying RSV is "the most important" viral respiratory pathogen is pretty strong. Influenza is also pretty serious in these groups.

We have adapted the text to “a very important viral pathogen”

We would like to thank the reviewer for his editing of spelling and language, all comments have been adapted in the text.

Line 76, "have been" should be "were". We thank the reviewer for this attention to detail and have changed all following concerns.

Line 90, dr should be Dr Has been changed.

Line 269, should read "scale bar". Has been changed.

Line 402, I'm not sure why Figure 7 is in bold. Has been changed.

Round 2

Reviewer 1 Report

As summarized in my first review, this work aims at addressing whether RSV A and B reference strains used for the majority of laboratory experiments are representative of the currently circulating viruses. The authors compared 8 RSV A isolates and 4 RSV B isolates to the lab-adapted RSV A2 and RSV B1 strains. They used multiple cell lines to assess viral growth kinetics, infectious virus production, thermal stability, syncytia formation, neutralization by palivizumab and mucin mRNA expression. The authors conclude that the reference strains are significantly distinct to currently circulating viruses.

The authors have failed to address some key comments of my previous review, and as indicated before, I think that the conclusions raised are not well supported by data. Moreover, if reported in these conditions, there is risk of highly confusing the field. There are two pieces of data that are critical and still missing. These are a characterization of the antiviral response in the cell lines used and by the different strains, as well as a full characterization of the content of defective interfering particles in each virus isolate before and after passage. Given the way that the isolates were propagated, it is possible that defective immunostimulatory viral genomes were amplified in some but not all the isolates complicating the interpretation of the data and compromising the conclusions raised.

Author Response

We agree with the comment of this and other reviewers that some of the conclusions we made are too strong and not fully substantiated by the data presented in the manuscript, especially concerning differences between the patient-derived viruses and the reference strains. Such conclusions have either been removed throughout the manuscript or were adjusted, so that these are in agreement with our experimental data and the suggestions of the reviewers.

On the request of additional experiments, we think the current manuscript already contains a lot of experimental data, and we feel that in the previous review reports the amount of additional experiments was too substantial. Among others, it was suggested to do the following experiments

Infection with the clinical isolates on primary epithelial cells Evaluation of antiviral, innate responses in infected cells Full genome sequencing of primary isolates from patient material and in the passage used for experiments Evaluation of replication levels by qPCR in addition to the data we generated with evaluation of production of infectious virus. This would require repeating all infection experiments. characterization of the content of defective interfering particles in each virus isolate before and after passage

Furthermore, we understand the specific concern of the reviewer and have added the following text to the discussion:

These differences could be due to underlying genetic differences that could either be accounted to antiviral responses in the cells, or due to differences that could be accounted to changes in the way the virus replicates in the cells, such as the presence of defective interfering virus particles.

Additionally, we have added a comment in the discussion for future research to determine defective interfering particles as such:

This suggests that the production of infectious virus particles is either limited or faulty in HEp-2 cells for this isolate. A full characterization of the production of defective interfering particles could provide an explanation.

Lastly, we have adapted the conclusion of the paper to the following:

We have observed differences in characteristics between viruses which were isolated from children in one RSV season, such as in vitro viral replication kinetics in cell lines, infectious virus production, thermal stability, fusion capacity and neutralization-sensitivity. Also, mucin mRNA expression was assessed and revealed generally minor differences between the different clinical isolates. Furthermore, a comparison of the clinical isolates to the reference strains RSV A2 and RSV B1 reveal overall no major differences. Clinical isolates can replicate either faster and slower in cell lines, generally have a lower thermal stability, and have differences in fusion capacity and mucin expression.